# Clinical Epidemiology of Microinvasive Cervical Carcinoma in an Italian Population Targeted by a Screening Programme

**DOI:** 10.3390/cancers14092093

**Published:** 2022-04-22

**Authors:** Lauro Bucchi, Silvano Costa, Silvia Mancini, Flavia Baldacchini, Orietta Giuliani, Alessandra Ravaioli, Rosa Vattiato, Federica Zamagni, Paolo Giorgi Rossi, Cinzia Campari, Debora Canuti, Priscilla Sassoli de Bianchi, Stefano Ferretti, Fabio Falcini

**Affiliations:** 1Emilia-Romagna Cancer Registry, Romagna Cancer Institute (*IRCCS Istituto Romagnolo per lo Studio dei Tumori (IRST) Dino Amadori*), 47014 Meldola, Italy; lauro.bucchi@irst.emr.it (L.B.); flavia.baldacchini@irst.emr.it (F.B.); orietta.giuliani@irst.emr.it (O.G.); alessandra.ravaioli@irst.emr.it (A.R.); rosa.vattiato@irst.emr.it (R.V.); federica.zamagni@irst.emr.it (F.Z.); fabio.falcini@irst.emr.it (F.F.); 2Department of Gynaecology, Madre Fortunata Toniolo Hospital, 40141 Bologna, Italy; costa.silvano@libero.it; 3Epidemiology Unit, Azienda Unità Sanitaria Locale—IRCCS di Reggio Emilia, 42122 Reggio Emilia, Italy; paolo.giorgirossi@ausl.re.it; 4Cancer Screening Unit, Azienda Unità Sanitaria Locale—IRCCS di Reggio Emilia, 42122 Reggio Emilia, Italy; cinzia.campari@ausl.re.it; 5Department of Health, Emilia-Romagna Region, 40127 Bologna, Italy; debora.canuti@auslromagna.it (D.C.); psassoli@regione.emilia-romagna.it (P.S.d.B.); 6Department of Translational Medicine, University of Ferrara and Local Health Authority, 44121 Ferrara, Italy; stefano.ferretti@unife.it; 7Local Health Authority, 47121 Forlì, Italy

**Keywords:** cervical carcinoma, microinvasive, lymph node dissection, hysterectomy

## Abstract

**Simple Summary:**

According to this population-based study, 3750 patients living in the Emilia-Romagna Region (northern Italy) were registered with cervical carcinoma between 1995–2016, including 2942 eligible patients. The likelihood of stage IA cervical carcinoma (*n* = 876, 29.8%) did not change over time, decreased with increasing patient age, and was lower for patients with adenocarcinoma and grade 3–4 disease. Three hundred and fifty (40.0%) patients had a conservative treatment, 317 (36.2%) had hysterectomy, 197 (22.5%) had hysterectomy with lymph node dissection (LND), and 12 (1.4%) had a conservative treatment with LND. The proportion of hysterectomy decreased from 70.6% in 1995–1999 to 46.9% in 2011–2016. The likelihood of hysterectomy increased above the age of 40. Among screening-aged (25–64 years) patients, the likelihood of hysterectomy did not differ between screen-detected and non-screen-detected ones. Hysterectomy was increasingly combined with LND. High tumour grade was the strongest determinant of LND during hysterectomy.

**Abstract:**

(1) Background: This population-based study aimed at identifying the factors associated with the likelihood of detection of stage IA cervical carcinoma—versus the detection of stage IB through IV cervical carcinoma—and the patterns of surgical treatment. (2) Methods: Between 1995–2016, 3750 patients living in the Emilia-Romagna Region (northern Italy) were registered with cervical carcinoma, including 2942 eligible patients (median age, 53). Multivariate analysis was performed using binary logistic regression models. (3) Results: The likelihood of stage IA cervical carcinoma (*n* = 876, 29.8%) did not change over time, decreased with increasing patient age, and was lower for patients with adenocarcinoma and grade 3–4 disease. Three hundred and fifty (40.0%) patients had a conservative treatment, 317 (36.2%) had hysterectomy, 197 (22.5%) had hysterectomy with lymph node dissection (LND), and 12 (1.4%) had a conservative treatment with LND. The proportion of hysterectomy decreased from 70.6% in 1995–1999 to 46.9% in 2011–2016. The likelihood of hysterectomy increased above the age of 40. Among screening-aged (25–64 years) patients, the likelihood of hysterectomy did not differ between screen-detected and non-screen-detected ones. Hysterectomy was increasingly combined with LND. High tumour grade was the strongest determinant of LND during hysterectomy. (4) Conclusions: This study provided a multifaceted overview of stage IA cervical carcinoma over the last decades.

## 1. Introduction

The surgical treatment of stage IA (including stages IA1 and IA2) cervical carcinoma, commonly referred to as microinvasive cervical carcinoma, has evolved during the last few decades. The main topics that have emerged include the conservative management of the disease with less radical procedures for selected patients, lymph node staging, and fertility preservation [1]. More defined criteria have been established, in particular for the identification of patients at no or low risk of disease progression—and thus treatable with conservative surgery—and for the use of pelvic lymph node dissection (LND) as a standard of care. In summary, the trend has been towards providing patients with tailored treatment that avoids morbidity while maintaining oncologic safety. Guidelines from major agencies and associations now include more defined treatment options [2,3,4].

However, it remains unclear whether these therapeutic indications have been accepted by the medical community as a whole, but especially in routine clinical practice settings. In fact, there are not sufficient objective data documenting the current patterns of care, particularly with regard to the use of conservative treatments and LND. Tumour stage and treatment are not routinely recorded in many cancer registries worldwide, but especially in Europe. Consequently, population-based, unselected clinical data have been reported only anecdotally and, to our knowledge, only from the United States [5,6,7,8,9]. In Europe, for example, the current United Kingdom practice of lymph node assessment in stage IA cervical carcinoma has been established through a web-based survey of the members of the British Gynaecological Cancer Society [10]. The study has been presented by the authors as ‘the only evidence available on clinical practice in this specific disease entity’. Incidentally, the results have confirmed the diversity of practice patterns.

The quality assurance and monitoring activities that are associated with ongoing organised cervical screening programmes, however, may help overcome the lack of information. In particular, those cancer registries that cover populations targeted by a screening programme are often supplemented with some basic clinical variables. In certain instances, these are collected for all incident cervical cancers irrespective of patient age and screening experience (or detection mode). This is the case for a cervical cancer registry covering a regional Italian population, which enabled us to conduct a population-based study on the factors associated with disease stage and with treatment of stage IA cervical carcinoma.

## 2. Materials and Methods

### 2.1. Setting

The study was conducted in the Emilia-Romagna Region, a large administrative region of northern Italy. On January 1, 2021, the total female population was 2,287,713. Since 1997, women aged 25–64 years have been targeted by a triennial cervical screening programme. Since 2015, the Pap smear has gradually been replaced with HPV testing in the age range 30–64 years. Women are invited with a personal letter to attend the district screening centres. The screening tests are taken by trained midwives. Colposcopy assessment for abnormal screening results and treatment of screen-detected lesions are performed at selected clinics of the Italian National Health Service.

Several previous papers have reported on the epidemiologic surveillance [11,12,13] and quality assurance activities for cytology [14], colposcopy [15], and surgical treatment of cervical intraepithelial neoplasia grade 2–3 [16]. Thus far, the average annual proportion of invited women aged 25–64 years undergoing screening has been 17.8%, corresponding to 53.4% on a triennial basis. Quality assurance initiatives are under the responsibility of the regional Department of Health and have a high degree of centralisation.

In 2008, a regional HPV vaccination campaign was commenced. The first cohorts offered the vaccine were born in 1996 and 1997 [12].

### 2.2. Rationale and Objectives

The primary objectives of the study were to identify the registered patient and disease characteristics associated with (1) the likelihood of detection of stage IA versus stage IB through IV cervical carcinoma and (2) the patterns of surgical treatment of stage IA patients. Secondary objectives were to determine the prevalence of positive pelvic lymph nodes among stage IA patients undergoing LND and to evaluate the 5-year overall survival.

### 2.3. Data

Copies of the records of invasive cervical carcinoma (International Statistical Classification of Diseases and Related Health Problems, 10th revision, code C53) [17] cases for the years 1995–2016 were obtained from the Emilia-Romagna cervical cancer registry, a joint initiative between the six accredited cancer registries that cover one or more districts of the region. More information on the registration process is reported elsewhere [12].

FIGO stage IA was defined as a depth of stromal invasion ≤5 mm and a horizontal extension ≤7 mm and was further subdivided into stage IA1 (depth of stromal invasion, <3 mm) and stage IA2 (depth of stromal invasion, ≥3 mm and ≤5 mm). This classification was taken from the 6th edition of the American Joint Committee on Cancer Staging Manual [18].

Conservative treatment was defined as uterus-preserving surgery (conization and trachelectomy, including radical trachelectomy). Hysterectomy included both simple and radical hysterectomy, regardless of the surgical approach.

Tumour grade was classified as well differentiated (grade 1), moderately differentiated (grade 2), poorly differentiated (grade 3), or undifferentiated (grade 4) [19].

The place of birth was classified into low- or high-emigration country based on criteria from the Italian National Institute of Statistics.

### 2.4. Statistical Methods

Since only part of patients were screening-aged, age and screening experience were combined into a single variable as follows: age <40 years; age 40–64 years, screen-detected; age 40–64 years, non-screen-detected; and age ≥65 years.

For univariate comparisons of patient characteristics, the Pearson chi-square test and the chi-square test for trend were used.

The independent determinants of the likelihood of patients with cervical carcinoma being diagnosed with stage IA versus more advanced disease, and the independent determinants of the patterns of surgical treatment of stage IA patients were identified using backward stepwise binary logistic regression models.

Lymph node status was descriptively reported.

Five-year overall percent survival was calculated from the date of registration and was estimated using the Kaplan–Meier method. The survival comparisons were based on the log-rank test. The prevalence of patients with positive lymph nodes and the incidence of death events were too few to create valid multivariate models in order to identify the determinants of both.

All analyses were performed using the Stata statistical software package, version 15.1 (Stata Corporation, College Station, TX, USA).

## 3. Results

### 3.1. Case Series

The case series included 3750 patients. Table 1 provides their distribution by eligibility status, tumour stage and age group. There were 375 (10.0%) patients staged IA1, 103 (2.7%) IA2 and 398 (10.6%) IA not otherwise specified, for a total of 876 (23.4%). Two thousand and sixty-six (55.1%) patients were clinically or pathologically staged IB through IV. They included 64 (1.7%) patients with post-therapy stage. Overall, there were 2942 (78.5%) patients eligible for the study (median age, 53 years; range, 19–93 years). Excluded from analysis were 41 (1.1%) patients reported to have a clinically staged IA cervical carcinoma—an information that we considered inconsistent and thus inaccurate—and 767 (20.5%) patients with unknown-stage disease.

### 3.2. Likelihood of Tumour Stage IA Versus IB trough IV

The 876 patients staged IA accounted for 29.8% of the 2942 all-stage eligible patients. They had a median age of 45 years (range, 19–92 years). The median age of the 2066 patients with stage IB through IV cervical carcinoma was 58 years (range, 23–93 years).

Table 2 shows the factors significantly associated with the likelihood of stage IA versus stage IB through IV. In multivariate analysis, the likelihood decreased consistently with increasing patient age and was lower for screen-detected patients aged 40–64 years than for the whole age group <40 years old, in which (data not shown) only 242/524 or 46.2% patients had a screen-detected disease. Stage IA was almost 50% less common among patients with adenocarcinoma. The low likelihood of stage IA associated with the ‘other, unknown’ tumour type category indicates a concentration of advanced-stage, inoperable lesions. Tumour grade, too, was a strong inverse predictor of early-stage disease, despite the high prevalence of missing information. Noteworthy, the odds ratio for the detection of stage IA disease did not change over time.

### 3.3. Surgical Treatment of Stage IA Patients

The distribution of the 876 patients with stage IA cervical carcinoma by pattern of surgical treatment is shown in Table 3. Three hundred and fifty (40.0%) of them had a conservative treatment (without LND), 317 (36.2%) were treated with hysterectomy, and 197 (22.5%) with hysterectomy and pelvic LND. Only 12 patients (1.4%) underwent conservative treatment with LND. In univariate analysis, age/screening experience, health care district of residence, time period, tumour type, and tumour grade were significantly associated with the surgical strategy. In particular, the proportion of total patients treated with hysterectomy (with or without LND) decreased from 70.6% in the late 1990s to 46.9% in 2011–2016. Hysterectomy was more frequent for stage IA2 than stage IA1 disease—approximately 70% versus 50% patients—and was more often combined with LND.

In multivariate analysis, which is shown in Table 4, the steep increase in the likelihood of hysterectomy (with or without LND) at age 40 years or older was confirmed, with no differentiation between screen-detected patients aged 40–64 years and non-screen-detected ones. Compared with patients living in north-western Emilia, the residents of the Central Emilia health care district were 2.5-fold more likely to be treated with hysterectomy and, when this occurred, two-fold more likely to receive LND. The multiple logistic regression models also confirmed the downward time trend in the likelihood of hysterectomy and the opposite trend in combining hysterectomy with LND. Adenocarcinoma was not associated with an increased risk of hysterectomy, but adenocarcinoma patients undergoing hysterectomy received more often LND. The strongest determinants of using LND during hysterectomy, however, were a tumour grade 2 to 4 and a disease stage IA2.

In a subgroup analysis of all screening-aged patients (25–64 years), screen-detected ones were confirmed not to have a lower likelihood to be treated with hysterectomy than non-screen-detected ones. The odds ratio for hysterectomy with LND, too, was near the unity (data not shown).

### 3.4. Lymph Node Status of Stage IA Patients

Out of the total 209 patients who underwent LND, five (2.4%) had one or more positive lymph nodes.

### 3.5. Survival of Stage IA Patients

The 5-year overall percent survival probability was 96.0% for stage IA1 patients and 96.0% for stage IA2 patients (logrank test; *p*-value, 0.362). In the whole population of total stage 1A patients, the number of deaths within 5 years of follow-up was 42, for a 5-year overall survival of 95.0% (95% CI, 93.2–96.3%).

In Table 5, the 5-year overall survival probability by tumour stage and treatment modality is shown. Among stage IA1 patients as well as patients staged IA without further specifications, the prognosis was significantly different between treatment groups, with better outcomes being observed in patients who received a conservative treatment compared with patients in the two hysterectomy groups. In a further analysis of patients with high-grade disease (*n* = 63), the two groups conservatively treated were pooled, and the same was done for the two hysterectomy groups. No significant difference in 5-year overall survival was found (data not shown).

None of the strata of patients defined by tumour stage (IA1, IA2, and IA not otherwise specified) and treatment modality (conservative, hysterectomy) showed significant improvements in 5-year overall survival across the time periods 1995–1999, 2000–2004, 2005–2010, and 2011–2016 (data not shown).

## 4. Discussion

To the best of our knowledge, this is one of the largest population-based clinical studies on stage IA cervical carcinoma patients ever published worldwide and the first from Europe. The results offer an essential but multifaceted overview of prevalence, treatment, and outcome of the disease in a defined population over the last decades.

The inverse association of glandular morphology and moderate-high tumour grade with the likelihood of patients with cervical carcinoma being diagnosed with a stage IA disease was expected. Less expected was the finding that early-stage disease was less likely among screen-detected patients aged 40–64 years than in the whole age group <40 years old, in which more than half patients were not diagnosed with cervical cancer through the screening programme. The most reasonable explanation is that they were diagnosed in the opportunistic screening setting, where the higher frequency of examinations and the higher degree of diagnostic intensity prevent more early carcinomas to progress to an advanced stage.

The absence of a significant down-staging over two decades, too, may appear surprising, in a population partly targeted by a screening programme. It must be considered, however, that the programme has led to an equal decrease in incidence of both early- and advanced-stage disease resulting in no effects on stage distribution. These trends are described is another paper [13].

The use of conservative treatments has been increasingly reported in the clinical literature [20,21,22,23]. On the basis of our results, we can confirm that this trend reflects a true increase at the population level. The proportion of patients treated with hysterectomy decreased from 70.6% in the late 1990s to 46.9% in 2011–2016. This finding was confirmed by multivariate analysis. It is interesting to note that, in the trend towards increasing use of conservative treatments, the association of LND remained an almost completely disregarded option.

Some of the factors associated with the type of treatment, too, were unexpected. This is especially the case for screening experience. Between 40 and 64 years of age, we found that screen-detected and non-screen-detected patients were treated according to the same surgical approach. From many points of view, this equality of management must be welcomed. Screen-detected patients, however, are theoretically referred to reference clinical centres selected by screening providers based on clinical quality criteria. A more conservative attitude to treating early invasive cancers is expected to be one of these—if not the main one. Our finding suggests that this may not be the case. As a consequence, referral practices, clinical counselling, patients’ compliance, and degree of multidisciplinary interaction [24] in the local screening programme warrant an audit.

Another unforeseen result was that the use of both hysterectomy and LND was more likely in the central health care district. This is characterised by the presence of highly qualified centres for gynaecologic oncology. In fact, while the decision to perform LND may be considered to reflect a higher level of surgical experience and competence, the use of hysterectomy indicates a problem of inappropriateness in many instances. We believe that the greater presence of private hospitals and clinics in the central health care district may be a factor in this apparently conflicting pattern.

We have previously demonstrated that screening-aged patients diagnosed with squamous carcinoma and adenocarcinoma in the study area between 1997 and 2012 have experienced a survival improvement of the same relative magnitude due to a phenomenon of downstaging within broad stage categories [25]. The current study draws attention to the fact that downstaging remains less effective for glandular lesions. In the whole case series, we found that the glandular tumour type was less likely to be diagnosed at an early stage. This was mostly accounted for by the pool of screen-detected cancers, in which the proportion of early-stage lesions was considerably lower among adenocarcinomas than squamous carcinomas (data not shown) reflecting the poor sensitivity of the Pap smear for microinvasive adenocarcinoma [26].

With respect to the prevalence of lymph node positivity, our finding of an overall figure of 2.4% was intermediate in the range reported in the literature, i.e., 0.5% to 7% [27,28]. However, despite the considerable size of this case series, patients with positive lymph nodes were too few in number to allow a robust estimate.

In 5-year overall survival analysis, patients conservatively treated exhibited a significantly better prognosis than patients treated with hysterectomy without LND and with LND, which indicates the safety of conservative surgical approaches. The difference in survival outcomes may result from post-hysterectomy morbidity, often associated with advanced age and medical comorbidities. However, a factor likely to contribute to the survival gap is a selection bias, due to the fact that conservative surgery is more often performed in patients at low, or very low, risk of disease progression.

Concerning patient survival, we also have to mention that external comparisons of our findings with other published data warrant caution. The observed 5-year overall survival probabilities were slightly lower than those generally reported by the Surveillance, Epidemiology, and End Results Program of the U.S. National Cancer Institute [7,29,30,31]. Simple comparisons between survival probabilities reported in different articles, however, are not adjusted for differences in methods and in distribution by major prognostic factors. In particular, the median patient age in this case series was as high as 45 years. It must be noted that, in an international study using standardised methods, the 5-year net survival from all-stage cervical carcinoma was higher in Italy than in the U.S. [32].

The major strength of this study lies in the fact that a series of registered cancer cases has inherently a large multicentre basis. An associated advantage is that a cancer registry-based series of patients has an unselected composition, which has probably allowed some unforeseen situations to surface. Cancer registries, conversely, are limited in the type and detail of data captured. In this study, in particular, we had not access to data on the HPV status of patients, on patients treated with radiotherapy, or on the type of hysterectomy and conservative treatment. Moreover, we could not evaluate the impact of the presence of lymphovascular emboli on treatment decisions nor could we distinguish between conization and trachelectomy (including radical trachelectomy).

## 5. Conclusions

The key findings of this study include the following: during the study period, the likelihood of tumour stage IA did not increase; glandular morphology and moderate-high tumour grade were inversely associated with the likelihood of early-stage disease; there was a downward time trend in the use of hysterectomy; unexpectedly, the surgical approach did not differ between screen-detected and non-screen-detected patients; the use of LND during hysterectomy increased over time; and high tumour grade was the strongest determinant of this surgical approach.

## Figures and Tables

**Table 1 cancers-14-02093-t001:** Distribution of the original registry-based series of cervical carcinoma patients by eligibility status, tumour stage, and age group. Emilia-Romagna Region, Italy (1995–2016).

Patient Eligibility Status and Tumour Stage	Patient Age (Years)			Total
	<40	40–64	≥65	
Eligible				
Pathologic stage IA1	135 (21.8)	202 (11.1)	38 (2.9)	375 (10.0)
Pathologic stage IA2	22 (3.5)	71 (3.9)	10 (0.8)	103 (2.7)
Pathologic stage IA NOS	129 (20.8)	215 (11.9)	54 (4.1)	398 (10.6)
Subtotal	286 (46.1)	488 (26.9)	102 (7.7)	876 (23.4)

Clinical or pathologic stage IB through IV	224 (36.1)	1028 (56.7)	750 (56.9)	2002 (53.4)
Any post-therapy stage	14 (2.3)	45 (2.5)	5 (0.4)	64 (1.7)
Subtotal	238 (38.4)	1073 (59.2)	755 (57.3)	2066 (55.1)

Subtotal	524 (84.5)	1561 (86.1)	857 (65.0)	2942 (78.5)
Not eligible				
Clinical stage IA1	3 (0.5)	4 (0.2)	4 (0.3)	11 (0.3)
Clinical stage IA2	0 (0.0)	3 (0.2)	0 (0.0)	3 (0.1)
Clinical stage IA NOS	3 (0.5)	5 (0.3)	19 (1.4)	27 (0.7)
Subtotal	6 (1.0)	12 (0.7)	23 (1.7)	41 (1.1)

Unknown stage/patient untreated	4 (0.6)	6 (0.3)	32 (2.4)	42 (1.1)
Unknown stage/missing data	86 (13.9)	233 (12.9)	406 (30.8)	725 (19.3)
Subtotal	90 (14.5)	239 (13.2)	438 (33.2)	767 (20.5)

Subtotal	96 (15.5)	251 (13.9)	461 (35.0)	808 (21.5)
Total	620 (100.0)	1812 (100.0)	1318 (100.0)	3750 (100.0)

NOS, not otherwise specified. Numbers in parentheses are column percentages.

**Table 2 cancers-14-02093-t002:** Univariate and multivariate association of registered patient and disease characteristics with the likelihood of stage IA versus stage IB through IV cervical carcinoma. Emilia-Romagna Region, Italy (1995–2016).

Characteristic	Number	Tumour Stage		*p*-Value *	Odds Ratio for Stage IA (95% CI)
		IA (*n* = 876)	IB through IV (*n* = 2066)		
Age (years) and screening experience				<0.001	
<40	524	286 (54.6)	238 (45.4)		Reference category
40–64, SD	596	292 (49.0)	304 (51.0)		0.71 (0.55–0.93)
40–64, non-SD	965	196 (20.3)	769 (79.7)		0.21 (0.17–0.28)
≥65	857	102 (11.9)	755 (88.1)		0.11 (0.09–0.15)
Place of birth				0.435	Variable removed
Low-emigration country	2623	775 (29.5)	1848 (70.5)		
High-emigration country	319	101 (31.7)	218 (68.3)		
Health care district				0.006	Variable removed
North-western Emilia	1254	354 (28.2)	900 (71.8)		
Central Emilia	677	235 (34.7)	442 (65.3)		
Romagna	1011	287 (28.4)	724 (71.6)		
Time period of diagnosis				0.001	Variable removed
1995–1999	809	262 (32.4)	547 (67.6)		
2000–2004	782	243 (31.1)	539 (68.9)		
2005–2010	726	226 (31.1)	500 (68.9)		
2011–2016	625	145 (23.2)	480 (76.8)		
Tumour type				<0.001	
Squamous carcinoma	2325	738 (31.7)	1587 (68.3)		Reference category
Adenocarcinoma	485	126 (26.0)	359 (74.0)		0.53 (0.41–0.70)
Other, unknown	132	12 (9.1)	120 (90.9)		0.24 (0.13–0.47)
Tumour location				0.017	Variable removed
Exocervical	547	143 (26.1)	404 (73.9)		
Endocervical	1056	345 (32.7)	711 (67.3)		
Unknown	1339	388 (29.0)	951 (71.0)		
Tumour grade				<0.001	
1	212	84 (39.6)	128 (60.4)		Reference category
2	588	93 (15.8)	495 (84.2)		0.27 (0.18–0.40)
3–4	807	63 (7.8)	744 (92.2)		0.14 (0.09–0.21)
Unknown	1335	636 (47.6)	699 (52.4)		1.33 (0.95–1.86)

CI, confidence interval; SD, screen-detected. Numbers in parentheses are row percentages. Odds ratios were from a binary logistic regression model (backward stepwise selection of variables). * For the Pearson chi-square test and the chi-square test for trend (time period of diagnosis).

**Table 3 cancers-14-02093-t003:** Univariate association of registered patient and disease characteristics with the pattern of treatment of stage IA cervical carcinoma. Emilia-Romagna Region, Italy (1995–2016).

Characteristic	Number	Pattern of Treatment				*p*-Value *
		Conservative without LND (*n* = 350)	Conservative with LND (*n* = 12)	Hysterectomy without LND (*n* = 317)	Hysterectomy with LND (*n* = 197)	
Age (years) and screening experience						<0.001
<40	286	193 (67.5)	3 (1.0)	51 (17.8)	39 (13.6)	
40–64, SD	292	89 (30.5)	6 (2.1)	122 (41.8)	75 (25.7)	0.378 †
40–64, non-SD	196	48 (24.5)	2 (1.0)	90 (45.9)	56 (28.6)	
≥65	102	20 (19.6)	1 (1.0)	54 (52.9)	27 (26.5)	
Place of birth						0.156
Low-emigration country	775	309 (39.9)	10 (1.3)	289 (37.3)	167 (21.5)	
High-emigration country	101	41 (40.6)	2 (2.0)	28 (27.7)	30 (29.7)	
Health care district						0.001
North-western Emilia	354	161 (45.5)	8 (2.3)	123 (34.7)	62 (17.5)	
Central Emilia	235	71 (30.2)	2 (0.9)	91 (38.7)	71 (30.2)	
Romagna	287	118 (41.1)	2 (0.7)	103 (35.9)	64 (22.3)	
Time period of diagnosis						<0.001
1995–1999	262	73 (27.9)	4 (1.5)	127 (48.5)	58 (22.1)	
2000–2004	243	101 (41.6)	1 (0.4)	96 (39.5)	45 (18.5)	
2005–2010	226	104 (46.0)	2 (0.9)	60 (26.5)	60 (26.5)	
2011–2016	145	72 (49.7)	5 (3.4)	34 (23.4)	34 (23.4)	
Tumour type						0.023
Squamous carcinoma	738	292 (39.6)	9 (1.2)	283 (38.3)	154 (20.9)	
Adenocarcinoma	126	52 (41.3)	3 (2.4)	30 (23.8)	41 (32.5)	
Other, unknown	12	6 (50.0)	0 (0.0)	4 (33.3)	2 (16.7)	
Tumour location						0.066
Exocervical	143	52 (36.4)	6 (4.2)	50 (35.0)	35 (24.5)	
Endocervical	345	145 (42.0)	2 (0.6)	121 (35.1)	77 (22.3)	
Unknown	388	153 (39.4)	4 (1.0)	146 (37.6)	85 (21.9)	
Tumour grade						<0.001
1	84	40 (47.6)	1 (1.2)	30 (35.7)	13 (15.5)	
2	93	18 (19.4)	2 (2.2)	30 (32.3)	43 (46.2)	
3–4	63	21 (33.3)	0 (0.0)	20 (31.7)	22 (34.9)	
Unknown	636	271 (42.6)	9 (1.4)	237 (37.3)	119 (18.7)	
Tumour stage						<0.001
IA1	375	174 (46.4)	4 (1.1)	129 (34.4)	68 (18.1)	
IA2	103	26 (25.3)	4 (3.9)	26 (25.2)	47 (45.6)	
IA NOS	398	150 (37.7)	4 (1.0)	162 (40.7)	82 (20.6)	

LND, lymph node dissection; NOS, not otherwise specified; SD, screen-detected. Numbers in parentheses are row percentages. * For the Pearson chi-square test. † Test for the difference between screen-detected and non-screen-detected patients aged 40–64 years.

**Table 4 cancers-14-02093-t004:** Multivariate association of the registered patient and disease characteristics with the likelihood of hysterectomy—versus conservative treatment—and the likelihood of hysterectomy with—versus without—lymph node dissection in the treatment of stage IA cervical carcinoma. Emilia-Romagna Region, Italy (1995–2016).

Characteristic	Odds Ratio (95% CI) for Hysterectomy * Versus Conservative Treatment	Odds Ratio (95% CI) for Hysterectomy with LND Versus without LND
Age (years) and screening experience		Variable removed
<40	1.00 (reference category)	
40–64, SD	5.35 (3.67–7.80)	
40–64, non-SD	6.57 (4.26–10.14)	
≥65	8.12 (4.58–14.37)	
Place of birth		Variable removed
Low-emigration country	1.00 (reference category)	
High-emigration country	1.59 (0.97–2.61)	
Health care district		
North-western Emilia	1.00 (reference category)	1.00 (reference category)
Central Emilia	2.51 (1.62–3.88)	2.00 (1.20–3.33)
Romagna	1.83 (1.18–2.86)	1.05 (0.60–1.87)
Time period of diagnosis		
1995–1999	1.00 (reference category)	1.00 (reference category)
2000–2004	0.51 (0.34–0.78)	0.95 (0.57–1.57)
2005–2010	0.42 (0.27–0.64)	2.28 (1.35–3.84)
2011–2016	0.33 (0.20–0.54)	2.47 (1.30–4.71)
Tumour type	Variable removed	
Squamous carcinoma		1.00 (reference category)
Adenocarcinoma		2.21 (1.25–3.93)
Other, unknown		0.71 (0.12–4.28)
Tumour grade		
1	1.00 (reference category)	1.00 (reference category)
2	2.77 (1.34–5.74)	3.95 (1.60–9.71)
3–4	1.68 (0.78–3.63)	3.60 (1.35–9.60)
Unknown	1.09 (0.65–1.82)	2.00 (0.91–4.39)
Tumour stage		
IA1	1.00 (reference category)	1.00 (reference category)
IA2	1.55 (0.90–2.67)	3.77 (2.04–6.95)
IA NOS	1.98 (1.32–2.98)	1.12 (0.69–1.84)

CI, confidence interval; LND, lymph node dissection; NOS, not otherwise specified; SD, screen-detected. Odds ratios were from two binary logistic regression models (backward stepwise selection of variables). Tumour location (exocervical, endocervical, unknown) was removed from both models as non-significantly contributing to their likelihood. * With or without LND.

**Table 5 cancers-14-02093-t005:** Five-year overall percent survival (with 95% confidence interval) of registered patients with stage IA cervical carcinoma, by tumour stage and pattern of treatment. Emilia-Romagna Region, Italy (1995–2016).

Tumour Stage	Pattern of Treatment				*p*-Value *
	Conservative without LND	Conservative with LND	Hysterectomy without LND	Hysterectomy with LND	
IA1	98.1 (94.3–99.4)	100.0 (NC)	95.9 (90.4–98.3)	90.3 (79.6–95.6)	0.033
IA2	87.4 (65.6–95.8)	100.0 (NC)	100.0 (NC)	97.9 (85.8–99.7)	0.403
IA NOS	95.9 (91.2–98.2)	NC (NC)	93.0 (87.7–96.1)	90.9 (81.9–95.6)	0.027

LND, lymph node dissection; NC, not calculable; NOS, not otherwise specified. Overall survival was calculated from the date of registration and was estimated using the Kaplan–Meier method. * For the log-rank test.

## Data Availability

The anonymised dataset used in this study is available on request from the corresponding author.

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
