# Peer review of "Clinical Epidemiology of Microinvasive Cervical Carcinoma in an Italian Population Targeted by a Screening Programme"

_cancers, 2022, doi:10.3390/cancers14092093_

Round 1

Reviewer 1 Report

Very interesting study and article.  But major gaps in the literature review.

Introduction:

  • You didn't cite the European and American Guidelines: ESGO and NCCN. In the Guidelines there are clear indication on how to treat all stages, including in case of preservation of fertility.
  • Numerous prospective studies have been conducted on LND, such as Sentinel 1 and 2. Others are going such as Sentinel 3, SentiX and Phoenix.
  • Preservation of fertility: we didn't cite the recent prospective study ConCerv nor the published meta-analyzes.

M&M:

  • Please, precise the FIGO stage used
  • Line 109-117 and Table 1 are to be put in the Results

Results:

  • Line 164-165 and table 3: please explain. Conservative = trachelectomy/radical trachelectomy or conisation? Hysterectomy and pelvic LND = simple hysterectomy or radical hysterectomy?

Discussion:

  • Please, comment the impact of HPV vaccination.
  • Concerning the survival, please discuss the impact of adjuvant treatments, probably added in case of positive lymph node.

Author Response

Introduction: -

–   You didn't cite the European and American Guidelines: ESGO and NCCN. In the Guidelines there are clear indication on how to treat all stages, including in case of preservation of fertility.

–   Numerous prospective studies have been conducted on LND, such as Sentinel 1 and 2. Others are going such as Sentinel 3, SentiX and Phoenix.

–   Preservation of fertility: we didn't cite the recent prospective study ConCerv nor the published meta-analyzes.

Cumulative response: I acknowledge that the original first paragraph of the Introduction section provided an outdated picture of the treatment of microinvasive cervical carcinoma. The criteria for uterus-preserving surgery and lymph node assessment are now more established than in the past, thanks to recent guidelines from major agencies. I have modified the paragraph (lines 48-57) as well as the references (#1-5). Also, I have modified the first lines of the second paragraph, in order to harmonize it with the initial one (lines 58-60).    

M&M:

Please, precise the FIGO stage used.

Response: In the cervical cancer registry that formed the basis for the study, tumour stage was originally classified using the 6th edition of the TNM staging system developed by the UICC and the AJCC (lines 112-114). I have provided the reference (#18). However, since this FIGO staging is also reported in this book and is perfectly equivalent to the TNM staging system, the authors preferred to use the FIGO nomenclature (Stage IA, IA1, IA2, etc.).

Line 109-117 and Table 1 are to be put in the Results

Response: The former lines 109-117 are now placed in the Results section (current lines lines 144-153) where they form the first paragraph (3.1. Case series). I also agree that Table 1 should be moved to the Results section. However, I hope this will be done by the Editorial Office. This highly-formatted Word file is too difficult to handle for me.       

Results:

Line 164-165 and table 3: please explain. Conservative = trachelectomy/radical trachelectomy or conisation? Hysterectomy and pelvic   LND = simple hysterectomy or radical hysterectomy?

Response: The cancer registry data do not allow more detailed classifications. ‘Conservative treatment’ includes any uterus-preserving surgery (conisation and trachelectomy, including radical trachelectomy). Similarly, the authors are unable to distinguish between simple hysterectomy and radical hysterectomy. In the Methods section, this is now pointed out (lines 115-117). Also, these limitations are acknowledged in the Discussion section (lines 321-325).

Discussion:

Please, comment the impact of HPV vaccination.

Response: HPV vaccination had no impact on cervical cancer experience of study patients. In this revised version, I have informed the reader that the regional HPV vaccination campaign was commenced in 2008 starting from girls in the 1996 and 1997 birth cohorts (lines 95-96). It clearly appears that these girls reached the age of screening (25 years) in 2020 and after, whereas the study covered the years 1995-2016.

Concerning the survival, please discuss the impact of adjuvant treatments, probably added in case of positive lymph node.

Response: The prognostic impact of treatment modalities in patients with positive lymph nodes was impossible to evaluate because, among the 209 patients undergoing lymph node dissection, only five were node-positive. This now pointed out in the Results section (lines 216-218).

Reviewer 2 Report

The authors attempted to identify the factors associated with the likelihood of detection of micro-invasive cervical cancer. This manuscript is well written and well organized. My minor comments are listed below.

Lines 104-106

Please cite previous studies.

Table 2

What is grade 3-4 cervical cancer? Please clarify and cite the previous studies for better understanding.

Table 3

Please clarify the number of women who were treated with radiotherapy.

The type of hysterectomy is unknown. Please add this point as a limitation of this study.

Author Response

The authors attempted to identify the factors associated with the likelihood of detection of micro-invasive cervical cancer. This manuscript is well written and well organized. My minor comments are listed below.

Response: Many thanks for this comment.

Lines 104-106

Please cite previous studies.

Response: In the (original) lines 104-106, the definitions of FIGO stage IA, stage IA1 and stage IA2 were provided. I have added a reference for this classification, that is, the AJCC Staging Manual (#18).

Table 2

What is grade 3-4 cervical cancer? Please clarify and cite the previous studies for better understanding.

Response: Grade 3 indicates a poorly differentiated cancer, and grade 4 an undifferentiated cancer. Grade 3-4 indicates the one or the other. In the Materials and Methods section, I have provided the definition of grade 1 to grade 4 (lines 118-119) and have referenced a classic paper describing the conventional grading system for cervical carcinoma (#19).

Table 3

Please clarify the number of women who were treated with radiotherapy.

Response: The authors had no access to data on patients treated with radiotherapy. This is now included among the acknowledged limitations of the study (line 322).

The type of hysterectomy is unknown. Please add this point as a limitation of this study.

Response: This is undoubtedly a limitation and is now acknowledged (line 322).

Reviewer 3 Report

In this manuscript, Bucchi L et al, have attempted to identify factors that are important for detection of stage 1A cervical carcinoma in comparison with stages 1B through IV and identify the patterns of surgical treatment. To this end, they used data collected as part of a screening program from cervical cancer patients belonging to Emilia-Romagna Region of Italy to perform multivariate analysis. They reported that proportion of hysterectomy decreased with the advancement in years and it was more common in patients above age of 40 years. Interestingly, use of hysterectomy was independent of the method used for patient identification (i.e., screening vs. non-screening). In addition, high tumor grade was associated with increased use of lymph node dissection during hysterectomy. This is the first study of its kind reporting data from Europe in this regard; however, there are few concerns with the study, which are listed below:

  • A large majority of cervical cancer cases (almost more than 95%) are caused by human papilloma virus infection. It would be interesting to know if the history of HPV infection had any impact on the modality of treatment given to the patients. Do authors have access to such information in their data set?
  • It will also be helpful if the authors could include information about patient survival following the different treatment modalities in patients with different stages. Despite not enough data available were there any interesting trends that the authors observed?
  • The authors are also recommended to comment in the discussion section if an additional benefit was observed with hysterectomy with higher tumor grade.

Author Response

In this manuscript, Bucchi L et al, have attempted to identify factors that are important for detection of stage 1A cervical carcinoma in comparison with stages 1B through IV and identify the patterns of surgical treatment. To this end, they used data collected as part of a screening program from cervical cancer patients belonging to Emilia-Romagna Region of Italy to perform multivariate analysis. They reported that proportion of hysterectomy decreased with the advancement in years and it was more common in patients above age of 40 years. Interestingly, use of hysterectomy was independent of the method used for patient identification (i.e., screening vs. non-screening). In addition, high tumor grade was associated with increased use of lymph node dissection during hysterectomy. This is the first study of its kind reporting data from Europe in this regard; however, there are few concerns with the study, which are listed below:

Response: I agree that the findings indicated by the Reviewer #2 are the most interesting ones.

A large majority of cervical cancer cases (almost more than 95%) are caused by human papilloma virus infection. It would be interesting to know if the history of HPV infection had any impact on the modality of treatment given to the patients. Do authors have access to such information in their data set?

Response: The HPV status was not among the standard registration variables. In this revised version, I have acknowledged that this was among the study limitations (line 321). It should be considered that, as is stated in the Materials and Methods section (lines 84-85), the transition to the HPV-based screening of women aged 30-64 years was initiated in 2015 and was very gradual. The patients in this case series were registered in the years 1995-2016 and, by implication, the overwhelming majority of them (if not all) were diagnosed with a Pap smear alone and not with a HPV test.

It will also be helpful if the authors could include information about patient survival following the different treatment modalities in patients with different stages. Despite not enough data available were there any interesting trends that the authors observed?

Response: I have added a Table (Table 5) showing the results of an overall survival analysis by treatment modality and tumour stage. Prognosis was significantly different between treatment groups, with better outcomes being observed in patients who received a conservative treatment compared with hysterectomy (lines 224-228). This finding is commented in the Discussion (lines 298-304). With respect to time trends, none of the strata of patients defined by tumour stage and treatment modality showed significant improvements in 5-year overall survival during the study period (lines 232-235). 

The authors are also recommended to comment in the discussion section if an additional benefit was observed with hysterectomy with higher tumor grade.

Response: I have added a statement on this. Among patient with high-grade disease, there was no significant difference in 5-year overall survival between the subgroup conservatively treated and the hysterectomy subgroups as a whole (lines 228-231).